# FedPCE: Federated Personalized Client Embeddings

## Abstract

Despite recent efforts, federated learning (FL) still faces performance challenges due to non-IID data distributions among clients. This distribution shift complicates the addition of new clients and the transfer of federally learned models to unseen data. Inspired by the adaptation ability of normalization layer parameters, we first demonstrate the effectiveness of models trained using FedBN when being adapted to so far unseen data. Specifically, we extend the adaptation method based on a visual analysis of the normalization layer feature vectors. We introduce Federated Personalized Client Embeddings (FedPCE), which utilizes local embeddings to capture the underlying structure of the normalization feature vectors and, by extension, the dataset. Our results show that FedPCE performs comparably to other common FL algorithms during both training and adaptation. Notably, FedPCE achieves this performance using only a fraction of the parameters during fine-tuning (32 parameters in our experiments) compared to other methods.

## 1 Introduction

The field of computer vision has progressed significantly in recent years, largely due to the emergence of deep learning. This rapid advancement of deep learning has been powered by the abundance of data available for training. Typically, the most effective setup in deep learning involves utilizing a single model on a centralized system that can access the complete training dataset.

However, collecting sufficient data on a central system to support effective deep learning is not always feasible. This is particularly true when dealing with personal or medical data, where sharing is restricted due to data privacy regulations. Federated learning (FL) has emerged as a solution to utilize decentralized data McMahan et al. (2017).

This decentralization of data poses several challenges. One of the most significant is that the data collected in this way is often not independently and identically distributed (non-IID) Li et al. (2022), Rieke et al. (2020). For example, this issue can arise when healthcare centers use different imaging devices, or when variations in user practices exist across devices. In general, non-IID data hinders the performance of FL Zhao et al. (2018). Moreover, problems may arise when a model, already trained on certain data, is applied to newly acquired data with a distribution that differs from the original training set. Under such circumstances, the model's performance is not guaranteed.

This scenario is common in the medical domain. Laws protecting patient privacy often make it difficult to gather data centrally, and establishing agreements with new medical centers to use their data can be a lengthy process. Some centers may be unable or unwilling to participate in a FL process, meaning their data can only be used to adapt the model and evaluate its performance.

For both challenges, there are suggested solutions involving the adaptation of normalization layers Li et al. (2021b), Li et al. (2016). We have observed that when using FedBN on artificially non-IID data, some feature vectors of normalization layers tend to cluster together. As shown in Figure 1, when clients with non-IID data are created using three different types of artificial degradation, the local normalization feature vectors of similar clients are closer to each other. We aimed to capture this similarity in a low-dimensional embedding space, leading to the introduction of FedPCE.

Our approach seeks to capture the underlying data structure through learnable embedding vectors. We modify the model architecture by replacing normalization layer parameters with feature vectors generated from an embedding vector through a Multi-Layer Perceptron (MLP). This allows us

to focus on the low-dimensional embeddings during fine-tuning, making it possible to use fewer parameters and training samples.

Our primary contribution is the introduction of a novel method in which all parameters of the normalization layers are generated from a local embedding vector. We further demonstrate that adapting these low-dimensional embedding vectors to new data is sufficient to achieve competitive performance, thereby reducing the dimensionality of the adaptation problem. This approach not only minimizes the number of parameters that need to be adjusted but also decreases the required data for adaptation, making it highly efficient for scenarios with limited data.

## 2 RELATED WORK

### 2.1 FEDERATED LEARNING

Federated learning (FL) was introduced by McMahan et al. (2017) by the development of Federated Averaging (FedAvg), which laid the foundations for distributed learning in scenarios where data privacy and communication efficiency is critical. Since then, FL has attracted significant attention due to its ability to collaboratively train machine learning models across decentralized data sources without sharing raw data.

A primary challenge in FL is the heterogeneity of data across clients, commonly referred to as non-IID (independently and identically distributed) data as this phenomenon makes the models trained using FedAvg suboptimal Li et al. (2020b), Zhao et al. (2018), Hsieh et al. (2020). Several approaches have been proposed to address this. FedProx Li et al. (2020a) extends FedAvg by introducing a proximal term to stabilize training on heterogeneous data. SCAFFOLD Karimireddy et al. (2020) mitigates the effects of client drift by correcting updates using control variates, FedMA Wang et al. (2020) uses matched averaging to better aggregate during global updates, MOON Li et al. (2021a) introduces contrastive learning to align model representations between the client and the server, further enhancing personalization, FedBS Idrissi et al. (2021) assigns weights to client model updates based on the local loss and switches to FedProx after a while, FedDNA Duan et al. (2021) weights the statistical parameters of the model differently and pFedLA Ma et al. (2022) uses server-side hypernetworks to personalize model aggregation for each client.

### 2.2 DOMAIN ADAPTATION

Domain adaptation aims to transfer knowledge from a source domain with abundant data to a target domain with limited data. Challenges arise because the data distribution in the target domain differs from that in the source domain. Li et al. (2016) has shown that batch norm layers can play an important role in adapting models to new domains. Lian et al. (2022) utilize scaling and shifting features to adapt models to new data, and in FedIN Feng et al. (2023) this approach is applied to improve FL. The significance of scale and shift is also evident in style transfer Dumoulin et al. (2016); Huang & Belongie (2017). Additionally, Feature-wise Modulation Layers (FiLM), which scale and shift feature vectors, have been shown to support synergistic learning on partially labeled region-based segmentations, as demonstrated by Billot et al. (2024).

## 3 METHOD

In this section, we briefly recall a widespread setting for personalized FL in Subsection 3.1 and normalization layers in Subsection 3.2. These concepts are necessitated for FedPCE, which is introduced here to promote decentralized learning with embeddings.

### 3.1 PERSONALIZED FEDERATED LEARNING

In this section, we briefly recall the concept of personalized FL and introduce the relevant notation. For this reason, we assume that no data can be shared with a central server or any other clients due to data privacy reasons. We consider a scenario with $N$ distinct clients (e.g., medical centers or user devices), each with access to a local dataset and corresponding labels, as well as their own

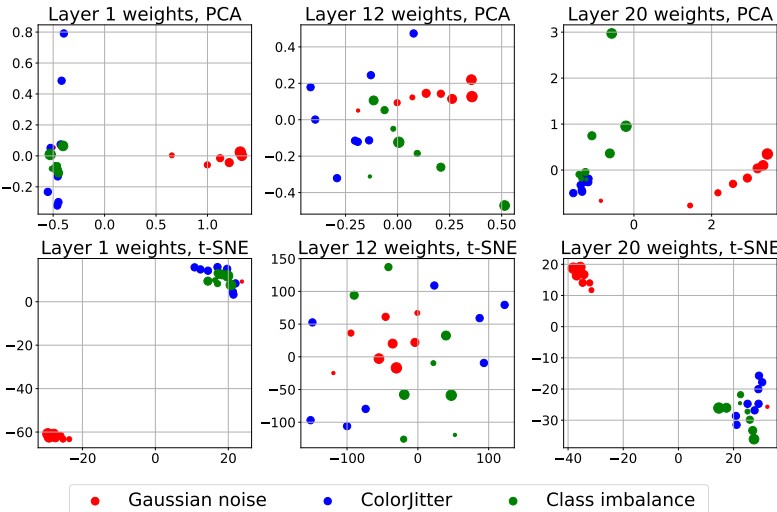

Figure 1: Visualisation of feature vectors of the first, $12^{th}$ and last normalization layers from the 24 collaborative training clients when using FedBN, first showing the vectors projected onto the first two dimensions of PCA, second using t-SNE. The data distribution at each client has been artificially altered by applying a degradation. Feature vectors corresponding to the same type of degradation appear closer to each other. Larger dots represent larger degradation levels for *Gaussian noise* and *Class imbalance*.

computational resources. Additionally, all clients are connected to a central server, where model aggregation occurs.

Let $C_1, \ldots, C_N$ represent the $N$ clients and denote the dataset available at client $C_n$ by $X_n = \{(x_n^\ell, y_n^\ell)\}_{\ell=1,\ldots,\ell_n}$, where each $(x_n^\ell, y_n^\ell)$ is a pair of input image and corresponding ground truth. The most widespread and easiest method to implement FL is Federated Averaging (FedAvg) McMahan et al. (2017). In FedAvg, all clients perform stochastic gradient descent using their local data, and after a fixed number of local iterations, send their model updates to a central server that averages the updates. In personalized FL, instead of updating all model parameters, we assume that certain parameters will remain local, meaning they will only be updated during local training. These parameters will neither be sent to the central server nor receive global updates. Both FedPer (Arivazhagan et al., 2019) and FedBN (Li et al., 2021b) can be understood from this point of view. In FedPer, the local parameters are the parameters of the last (few) layers, and in FedBN, they comprise the normalization layers. We denote the entity of model parameters at client $C_n$ by $\Omega_n$ and split them into global parts $\Omega_n^{gl} = \{\omega_n^i\}_{i \in I}$ and local parts $\Omega_n^{loc} = \{\omega_n^i\}_{i \in J}$, where $I$ and $J$ denote the index sets of the global and local parameters, respectively. This split is the same for each client.

In many cases, the objective function in FL can be framed as solving the following minimization problem:

$$\min_{\Omega_1, \ldots, \Omega_N} \sum_{n=1}^{N} |X_n| \cdot L(X_n; \Omega_n),$$

where $L$ represents the loss function, and $|X_n|$ is the number of data points at client $C_n$.

The training process proceeds as follows: all local model parameters are initialized identically. Each client $C_n$ then runs a gradient descent algorithm using its local data $X_n$ and model parameters $\Omega_n$ to minimize the local loss $L(X_n; \Omega_n)$. After a fixed number of iterations, clients send their global weight updates to the central server, which averages these updates to yield

$$(\omega_k^i)' = \frac{1}{N} \sum_{n=1}^{N} \omega_n^i \qquad \forall 1 \leq k \leq N, \, i \in I.$$

Crucially, local parameters are neither sent to the server nor updated during global model aggregation. This process alternates between local training and global parameter aggregation until a termination condition is met—in this case, aggregating the models for a specified number of times.

## 3.2 NORMALIZATION LAYERS

FedBN attempts to address the problem posed by data shift in FL by keeping the parameters of the normalization layers local. In the case of the most common normalization layers (e.g., Batch Norm (Ioffe & Szegedy, 2015), Instance Norm (Ulyanov et al., 2016), Layer Norm (Ba et al., 2016)), this refers to the scale and shift parameters. These were introduced by Ioffe & Szegedy (2015) to restore the representation power of the network after normalization. The action of these normalization layers is as follows for a feature vector $x$:

$$\hat{y} = \frac{x - \mathbb{E}[x]}{\sqrt{\text{Var}(x)}}, \tag{1}$$

$$y = \hat{y} \cdot \gamma + \beta, \tag{2}$$

where the expectation and variance are calculated along dimensions depending on the specific choice of normalization layer, and $\beta$ and $\gamma$ are learnable parameters. FedBN personalizes models by keeping the parameters $\beta$ and $\gamma$ local, i.e., these parameters are not shared among the clients.

## 3.3 LOCAL EMBEDDINGS

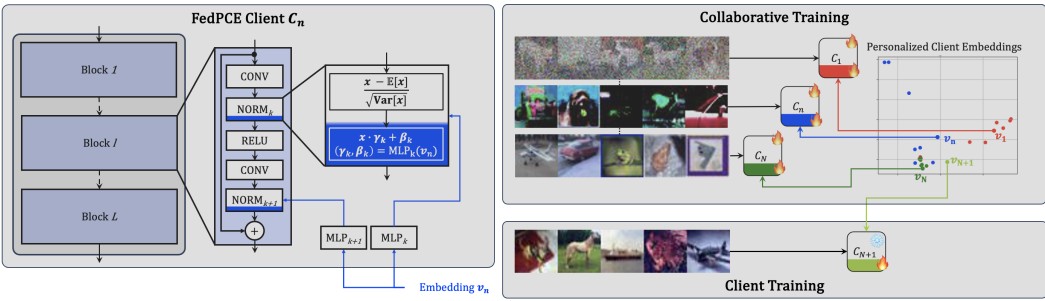

Figure 2: Left: In FedPCE, a client $C_n$ is personalized to the local data distribution by conditioning the scale $\gamma$ and shift $\beta$ vectors of all normalization layers onto a semantic embedding vector $v_n$. Top right: This embedding is learned during collaborative training of multiple clients that share all trainable parameters except the local embedding vectors $v_n$, $n = 1 \ldots N$. Bottom right: The semantic embedding space enables an effective personalization to new clients $C_{N+1}$ by only adapting its embedding vector $v_{N+1}$.

To distill the information encoded in the personalized shift and scale parameters depicted in Figure 1, we facilitate local embeddings. Using these embeddings, we drastically reduce the number of client-specific parameters and thereby simplify extending federated models to new clients.

Let there be $K$ normalization layers, and denote the shift and scale vectors at normalization layer $k$ by $\beta_k$ and $\gamma_k$, respectively. Usually $\beta_k$ and $\gamma_k$ are learnable parameters during federated training and are either globally shared (FedAvg) or locally personalized (FedBN). Instead of directly parameterizing $\beta_k$ and $\gamma_k$, we introduce a Multi-Layer Perceptron ($\text{MLP}_k$) in each normalization layer to predict suitable shift and scale parameters from a client specific embedding vector $v_n \in \mathbb{R}^E$, i.e.,

$$(\beta_k, \gamma_k)(v_n) = \text{MLP}_k(v_n). \tag{3}$$

Thus, each of the $K$ MLPs implements a nonlinear mapping from the embedding space $\mathbb{R}^E$ to the parameters space of the $k$th normalization layer $\mathbb{R}^{2 \dim(y_k)}$. Here $\dim(y_k)$ denotes the number of feature channels of the $k$th normalization layers' input. As a result, the rule (equation 2) of each normalization layer used in our approach changes to

$$y = \hat{y} \cdot \gamma_j(v_i) + \beta_j(v_i). \tag{4}$$

Each MLP consists of fully-connected layers with ReLU activation functions (Nair & Hinton, 2010) in between. The complexity of these MLPs is defined by the width and the number of hidden layers.

## 3.4 Decentralised learning with embeddings

We now describe the usage of embeddings, which involves first training the model using the clients available for FL—a process we refer to as collaborative training (Col. tr.)—and then adapting it to unseen data, which we will call client training (Cl. tr.).

*Collaborative training* is conducted using personalized FL, as outlined in Subsection 3.1. At each client $C_n$, only $\boldsymbol{v}_n$ serves as a local parameter, i.e., $\Omega_n^{loc} = \{\boldsymbol{v}_n\}$, while the remaining weights are global, denoted as $\Omega_n^{gl}$. Specifically, the weights of the MLPs mapping from the embedding to the normalization shifts and scales are global parameters. Thus, only an $E$ dimensional vector characterizes the personalization of each client. To initialize the embedding vector of each client for collaborative training, we set $\boldsymbol{v}_n = \boldsymbol{e}^{(n)}$ if $n \leq E$ and $\boldsymbol{0}$ otherwise. We experimented with different initialization strategies but did not observe any empirical difference.

The collaborative training enables the model to distill knowledge from the different local data distributions and represent it by local $\boldsymbol{v}_n$. Thereby, encoding semantic information in the associated embedding. Since the weights of the MLPs are global, the model collectively learns how to interpret this condensed information.

In the *client training* process, we utilize this encoded representation of the training data distribution to effectively personalize new clients to its associated data. To do so, we freeze all global parameters of the model and only fine-tune the embedding vector $\boldsymbol{v}_{N+1}$, as shown in the bottom right of Figure 2. Due to the low dimensionality of the embedding, only a small subset of the model's parameters have to be fine-tuned during client training, which reduces the number of labeled data samples required to effectively adapt the model to unseen clients.

# 4 Experiments

## 4.1 Experimental design and datasets

We now describe the experiments that will demonstrate the effectiveness of local embeddings in adapting a model to unseen data. The experiments simulate a scenario in which data centralization is not possible, and the data at some clients cannot be used for training; it can only be used to adapt the model for local usage. Therefore, we will generate a number of clients with artificially non-IID data distribution, use some of them for collaborative training, and then fine-tune the model on the remaining clients during client training.

To demonstrate the versatility of embeddings, we conduct experiments on three standard datasets:

- *CIFAR-10* and *CIFAR-100* Krizhevsky et al. (2009) are classification datasets, each consisting of 50,000 training and 10,000 validation images. Each of these RGB images is of size $32 \times 32$ and belongs to one of 10 or 100 classes, respectively.

- The third dataset, which we denote as *Digits*, is the union of four different public computer vision datasets, all containing images of digits (0-9). These are the *MNIST* (LeCun, 1998), *USPS* (Hull, 1994), *SVHN* (Netzer et al., 2011), and *SYN* (Roy et al., 2018) datasets. They contain 60,000, 7,291, 73,257, and 10,000 training images, respectively, and 10,000, 2,007, 26,032, and 2,000 validation images, respectively. The SVHN and SYN datasets consist of colored images, while MNIST and USPS contain grayscale images. For training, all images were resized to $32 \times 32$ using bilinear interpolation. We convert all grayscale images to color images.

## 4.2 Simulating non-IID data

For the Digits dataset, we set the total number of clients to be a multiple of 4, denoted as $4K$. For each dataset in Digits, we split the images uniformly into $K$ subsets. This results in $4K$ clients, simulating 4 different modalities, each with $K$ clients.

For CIFAR-10 and CIFAR-100, we apply three data degradation techniques to artificially alter the data distribution. These methods are as follows:

- *Gaussian noise*: We apply pixel-wise additive Gaussian noise to the images. For each client, the variance of the Gaussian noise is fixed, but the noise instance is randomly sampled every time. If there are $M$ total clients with *Gaussian noise* degradation, the variances used are $M$ points chosen linearly between 0.005 and 1.

- *ColorJitter*: Modeled after PyTorch's `ColorJitter`, this method adjusts the brightness, contrast, saturation, and hue of images. The extent of these adjustments is fixed for each client and determined as follows: For $M$ *ColorJitter* clients, we take $M$ points linearly spaced between 0.5 and 1.5 for brightness $(b_1, \ldots, b_M)$, contrast $(c_1, \ldots, c_M)$, and saturation $(s_1, \ldots, s_M)$, and between -0.5 and 0.5 for hue $(h_1, \ldots, h_M)$. We then randomly permute the values for each category separately, i.e. chose four random permutations of $(1, \ldots, M)$, $\varphi_1, \varphi_2, \varphi_3$ and $\varphi_4$. Then the $i^{th}$ client will be assigned the parameters $b_{\varphi_1(i)}$, $c_{\varphi_2(i)}$, $s_{\varphi_3(i)}$ and $h_{\varphi_4(i)}$ for brightness, contrast, saturation, and hue, respectively. The order of the change in brightness, contrast, saturation, and hue is applied in a random order.

- *Class imbalance*: When generating the clients corresponding to this degradation, the distribution of class labels is not uniform. If there are $2M$ *Class imbalance* clients, we choose $M$ values of $\alpha$ (logarithmically spaced between 0.1 and 10) and divide the entire set of images for these clients into $M$ subsets uniformly. Each $\alpha$ is paired with one subset, and the images of that subset are distributed using the Dirichlet distribution with the corresponding $\alpha$. This ensures that some clients have a fairly uniform class distribution (corresponding to a high $\alpha$ value), while others have a very uneven class distribution (corresponding to a low $\alpha$).

We set the total number of clients as a multiple of 6, denoted by $6M$. Each degradation type is assigned to $2M$ clients, ensuring an even number of *Class imbalance* clients. The dataset is uniformly divided into three parts, one for each degradation method. These parts are then further subdivided into $2M$ clients: uniformly for *Gaussian noise* and *ColorJitter*, and using the previously described method for *Class imbalance*. Training and validation images are partitioned separately but follow the same distribution. *Gaussian noise* and *ColorJitter* are applied to both the training and the validation images.

The data partitioning process, a random selection of *ColorJitter* and *Gaussian noise* parameters, *Gaussian noise* application, and data loading are performed in a reproducible manner, ensuring fair comparisons between models trained under the same conditions.

## 4.3 ARCHITECTURE

We will conduct all of our experiments using *ResNet-18* He et al. (2016). The models are implemented as described in He et al. (2016), with the following modifications. We use the 'CIFAR' version of ResNet, meaning the first convolution layer has a kernel size of 3, a stride of 1, and padding of 1, instead of the usual kernel size of 7, stride of 2, and padding of 3. We use instance normalization layers (Ulyanov et al., 2016) instead of the usual batch normalization Ioffe & Szegedy (2015), as we observed that instance normalization improves the performance of FedPCE. Additionally, we insert an extra normalization layer into the classification head, right before the final fully connected layer. This allows the embeddings to more directly influence the class predictions through the shift and scale vectors of the normalization layer, which is particularly helpful in the case of *Class imbalance*.

Because we only need to fine-tune the low-dimensional embedding vector in FedPCE during client training, the fine-tuning process requires training significantly fewer parameters. Although a ResNet-18 adjusted to FedPCE is 6.9% larger than ResNet-18 in terms of the number of parameters due to the additional MLPs (11.9M vs. 11.2M), fine-tuning it requires training only 32 parameters, compared to 10,624 for FedBN, representing a 332-fold decrease. To show that the performance does not stem from the extra number of parameters, we also conduct our experiments on a smaller version of ResNet and FedPCE, which we will denote by FedPCE(62). This is the same architecture as a ResNet-18 adjusted to FedPCE but we replace the original channel dimensions of the ResNet blocks of (64, 128, 256, 512) by (62, 124, 248, 496). FedPCE(62) has only an extra 600K parameters, a difference of 0.54% compared to ResNet-18.

We will compare the collaborative and client training of our models with four well-established FL methods: *FedAvg* (McMahan et al., 2017), *FedProx* (Li et al., 2020a), *FedPer* (Arivazhagan et al., 2019), and *FedBN* (Li et al., 2021b). In FedProx, the weight of the proximal loss term is set to

Table 1: Illustration of the accuracy of various FL algorithms on the CIFAR-10, CIFAR-100, and Digits datasets, for both collaborative and client training. The rightmost column displays the number of parameters optimized during the training phases for each method.

| | CIFAR-10 | | CIFAR-100 | | Digits | | # of parameters | |
|---|---|---|---|---|---|---|---|---|
| | Col. tr. (%) | Cl. tr. (%) | Col. tr. (%) | Cl. tr. (%) | Col. tr. (%) | Cl. tr. (%) | Col. tr. | Cl. tr. |
| FedAvg | $70.62 \pm 0.45$ | $52.13 \pm 1.65$ | $35.23 \pm 0.2$ | $24.71 \pm 0.66$ | $92.96 \pm 0.12$ | $92.15 \pm 0.69$ | 11.2M | - |
| FedProx | $70.54 \pm 0.27$ | $52.12 \pm 1.14$ | $35.07 \pm 0.28$ | $24.56 \pm 0.95$ | $93.04 \pm 0.12$ | $\mathbf{92.67 \pm 0.35}$ | 11.2M | - |
| FedPer | $67.96 \pm 0.42$ | $63.67 \pm 1.03$ | $24.97 \pm 0.36$ | $24.49 \pm 0.69$ | $92.93 \pm 0.14$ | $92.16 \pm 0.17$ | 11.2M | 5.1K |
| FedBN | $\mathbf{70.83 \pm 0.33}$ | $\mathbf{66.44 \pm 0.83}$ | $\mathbf{35.63 \pm 0.37}$ | $\mathbf{31.99 \pm 0.78}$ | $92.37 \pm 0.12$ | $89.60 \pm 0.46$ | 11.2M | 10.6K |
| FedPCE | $70.71 \pm 0.26$ | $65.67 \pm 0.82$ | $33.26 \pm 0.33$ | $27.97 \pm 0.91$ | $\mathbf{93.24 \pm 0.09}$ | $91.95 \pm 0.17$ | 12M | 32 |
| FedPCE(62) | $70.74 \pm 0.32$ | $65.77 \pm 1.06$ | $32.91 \pm 0.3$ | $26.87 \pm 1.08$ | $93.11 \pm 0.15$ | $91.75 \pm 0.19$ | 11.2M | 32 |

$\mu = 0.01$. In FedPer, the parameters of the last fully connected layer are kept local, while FedBN involves keeping the parameters of the normalization layers local. If a method has local parameters, we fine-tune those during client training. If it has no local parameters, we do not fine-tune the model during client training but use the validation sets of the fine-tuning clients to evaluate the model from collaborative training All other implementation details are the same across methods, as described in Section 5.

We conduct each experiment using 5-fold cross-validation. To implement this, we first partition the dataset among the clients, and then each client $C_n$ splits its local dataset $X_n$ into 5 folds. The partitioning of the local datasets and the selection of folds are carried out in a reproducible manner, ensuring that the training and validation sets at each client remain consistent. During each experiment, one fold serves as validation set, the rest of the dataset is used as training set, meaning at each site 80% of the data is used for training and 20% for validation.

## 5 NUMERICAL RESULTS

In this section, the numerical results of the experiments are presented and discussed along with an ablation study of the model's hyperparameters.

### 5.1 TRAINING DETAILS AND BENCHMARK RESULTS

In the main experiments, we use a total of 30 clients for CIFAR-10 and CIFAR-100, and 40 clients for Digits. In all cases, 80% of the clients are used for collaborative training, and the remaining 20% for client training. Unless stated otherwise, we use MLPs with 2 layers and a hidden layer dimension of 64. Local training is conducted using the Adam optimizer (Kingma & Ba, 2014) with an initial learning rate of $10^{-4}$, betas of $(0.5, 0.9)$, a weight decay of $10^{-4}$, and cosine learning rate scheduling with a minimum learning rate of $10^{-6}$. For FedPCE, the same optimizer and scheduler are used, except for the embeddings, where the initial learning rate is 0.1 with a minimum of $10^{-4}$, and for the MLP parameters, the initial learning rate is $10^{-2}$. We use a batch size of 64 and train for 50 local iterations between each global model aggregation. The same hyperparameters are used during client training, except for the starting learning rate for the embeddings which is $10^{-2}$. Collaborative training is run until the $1000^{th}$ global aggregation, while client training is done for 200 global aggregations. The training images are augmented by first padding with 4 pixels of value 0 on all sides, randomly cropping the image to $32 \times 32$, flipping the image horizontally with a probability of 0.25, rotating it by a degree uniformly sampled from $(-15°, 15°)$ with a probability of 0.25, and randomly erasing a rectangle of size between 16 and 256 pixels with a probability of 0.5.

In these experiments, we compare our method FedPCE, its smaller version, FedPCE(62), and the ResNet-18 architecture used in conjunction with four different FL methods: FedAvg, FedProx, FedPer, and FedBN. The results of the experiments are shown in Table 1.

The numerical results show that FedBN yields the highest accuracy scores for collaborative and client training for both CIFAR datasets, but it personalizes the largest number of parameters in client training. For CIFAR-10, our approach generates comparable results for collaborative and client training and even outperforms FedBN on the Digits dataset. Interestingly, FedAvg and FedProx achieved the highest accuracy scores on the Digits dataset for new clients, although both methods do not personalize.

Table 2: Results from collaborative and client training on the CIFAR-10 and CIFAR-100 datasets, organized by the degradation of clients. Displayed are the mean accuracies over the clients with a given degradation.

| CIFAR-10 | Gaussian noise | | ColorJitter | | Class imbalance | |
|---|---|---|---|---|---|---|
| | Col. tr. (%) | Cl. tr. (%) | Col. tr. (%) | Cl. tr. (%) | Col. tr. (%) | Cl. tr. (%) |
| FedAvg | 53.23± 0.3 | 14.45± 2.42 | 75.36± 0.79 | 64.24± 2.95 | 83.4 ± 0.76 | 77.63 ± 2.42 |
| FedProx | 53.4 ± 0.34 | 14.23± 2.78 | 75.11± 0.62 | 64.09± 1.74 | 83.3 ± 0.58 | 77.97 ± 0.94 |
| FedPer | 51.05± 0.42 | 46.19± 1.22 | 72.11± 0.94 | 67.39± 2.8 | 80.84± 0.49 | 77.39 ± 0.69 |
| FedBN | 53.27± 0.63 | 47.92± 0.84 | **76.01±0.83** | **70.87±1.98** | 83.37± 0.53 | **80.49±1.64** |
| FedPCE | **53.35±0.69** | **48.92±0.28** | 75.33± 0.48 | 68.49± 1.85 | **83.59±0.42** | 79.58 ± 1.36 |
| FedPCE(62) | 53.2 ± 0.77 | 48.29± 0.84 | 75.76± 0.67 | 69.42± 2.05 | 83.43± 0.43 | 79.58 ± 1.74 |

| CIFAR-100 | Gaussian noise | | ColorJitter | | Class imbalance | |
|---|---|---|---|---|---|---|
| | Col. tr. (%) | Cl. tr. (%) | Col. tr. (%) | Cl. tr. (%) | Col. tr.(%) | Cl. tr. (%) |
| FedAvg | **23.32±0.47** | 9.11± 1.29 | 35.43± 1.4 | 23.33± 0.85 | 46.88± 0.91 | 41.53 ± 0.72 |
| FedProx | 23.00± 0.8 | 9.27± 1.37 | 35.24± 0.72 | 22.88± 1.56 | **46.92±1.13** | 41.37 ± 0.78 |
| FedPer | 17.11± 0.3 | 18.05± 0.73 | 23.74± 0.88 | 21.26± 1.89 | 34.06± 0.87 | 34.08 ± 1.33 |
| FedBN | 23.07± 0.38 | 20.64± 0.67 | **37.01±0.74** | **30.13±1.26** | 46.75± 0.58 | **45.10±1.67** |
| FedPCE | 22.94± 0.39 | **22.25±0.88** | 33.38± 1.14 | 24.82± 1.39 | 43.40± 0.75 | 36.75 ± 1.49 |
| FedPCE(62) | 23.06± 0.47 | 22.15± 0.89 | 32.65± 0.81 | 22.15± 1.61 | 42.98± 1.19 | 36.23 ± 1.43 |

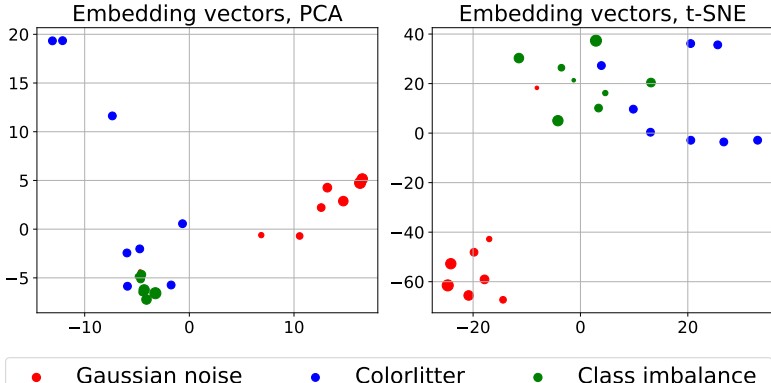

Figure 3: Visualization of the embedding vectors after collaborative training using the CIFAR-100 dataset. The first plot shows the vectors projected onto the first two dimensions of PCA, the second uses t-SNE. The embedding vectors of clients corresponding to the same degradation appear closer to each other. Larger dots represent larger degradation levels when applicable, i.e., for *Gaussian noise* and *Class imbalance*.

To investigate the performance drop of FedPCE on the CIFAR-100 dataset, we listed the stratified accuracies for the different degradations in Table 2. Here, we observe that our approach clearly outperforms all others for new clients degraded by *Gaussian noise* on CIFAR-10 and CIFAR-100. However, client training for *ColorJitter* and *Class imbalance* works best using FedBN. We argue that the performance drop of FedPCE for these degradations originates from the limited expressivity of the 32-dimensional embedding space, which cannot smoothly encode all relevant properties for CIFAR-100.

## 5.2 ABLATION STUDIES

Table 3: Ablation study for the embedding dimension.

| $E$ | 2 | 4 | 8 | 16 | 32 | 64 |
|---|---|---|---|---|---|---|
| Col. tr. (%) | 70.52± 0.27 | 70.85± 0.48 | 70.68± 0.21 | 70.67± 0.18 | 70.71± 0.26 | 70.82± 0.39 |
| Cl. tr. (%) | 65.18± 0.57 | 65.67± 0.62 | 66.10± 0.95 | 65.64± 1.08 | 65.67± 0.82 | 66.72± 0.79 |

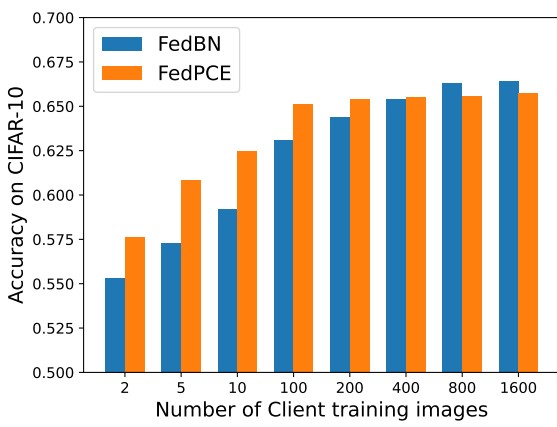

Figure 4: Performance of FedBN and FedPCE trained models when the number of available training images is restricted during client training.

Table 4: Ablation study for the number of clients.

| # clients | 12 | 30 | 60 | 90 |
|---|---|---|---|---|
| Col. tr. (%) | $73.06_{\pm 0.5}$ | $70.71_{\pm 0.26}$ | $68.52_{\pm 0.36}$ | $66.50_{\pm 0.31}$ |
| Cl. tr. (%) | $65.69_{\pm 0.4}$ | $65.67_{\pm 0.82}$ | $68.10_{\pm 0.96}$ | $68.20_{\pm 0.59}$ |

To better understand the effects and behavior of embeddings, we conduct a series of ablation studies concerning the embedding dimension, the number of clients, the structure of the MLPs, and the number of training images during client training. All the ablation experiments were conducted on CIFAR-10 using the training setup described in Subsection 5.1 unless specified otherwise.

First, we demonstrate that, due to the low dimensionality of the embedding space, a model trained with FedPCE can be fine-tuned effectively using a very small number of data points. In this experiment, we compare FedBN and FedPCE models that were collaboratively trained on CIFAR-10. During client training, we limit the number of available training images per client, ranging from as few as 2 images to the full dataset of 1600 images per client. The validation set remains consistent across all trials. The results are illustrated in Figure 4 and show that FedPCE consistently outperforms FedBN for a low number of available training samples. Its performance stabilizing after around 100 training samples. Beyond this point, the additional fine-tuning parameters of FedBN enable it to close the gap with FedPCE and eventually surpass it for more than 800 training samples. This demonstrates FedPCE's advantage in low-data regimes, while also highlighting the benefits of FedBN's more complex adaptation as the data volume increases.

Table 3 presents the results of the embedding dimension ($E$) ablation experiments conducted with FedPCE, using dimensions ranging from 2 to 64. The findings reveal that the performance of Fed-PCE is not significantly impacted by the embedding dimension. Strong results were achieved in both collaborative and client training even with smaller embedding sizes, indicating that the method remains robust across various embedding configurations.

Table 4 shows the results of experiments conducted with FedPCE across 12, 30, 60, and 90 clients. As anticipated in FL, collaborative training accuracy declines as the number of clients increases

Table 5: Ablation study for the dimension of the hidden layer.

| hidden dim | 16 | 64 | 256 |
|---|---|---|---|
| Col. tr. (%) | $70.66_{\pm 0.22}$ | $70.71_{\pm 0.26}$ | $71.17_{\pm 0.37}$ |
| Cl. tr. (%) | $65.47_{\pm 0.8}$ | $65.67_{\pm 0.82}$ | $66.50_{\pm 0.83}$ |

due to the increased data heterogeneity. However, client training accuracy exhibits an upward trend as the number of clients grows. This suggests that the model's MLPs become increasingly adept at interpreting and adapting to diverse data when exposed to a larger pool of clients during collaborative training.

Finally, Table 5 examines the effect of the MLPs' size on the performance of FedPCE. We evaluated 2-layer MLPs with hidden dimensions of 16, 64, and 256. While increasing the hidden dimension results in slight performance gains, the improvements are not substantial.

Despite the advantages of FedPCE presented above there are some minor limitations. For datasets with a large number of classes, the embeddings might not fully capture the complexity of the underlying data. Additionally, the introduction of MLPs increases model complexity and might lead to more difficult optimization.

## 6 CONCLUSION

In this work, we proposed to distill the implicit representations encoded in the parameters of personalized normalization layers in FL using embeddings. These embeddings are learned from multiple clients during collaborative training. Afterwards, the model can be easily extended to new clients with different data distributions by just fine-tuning the client's embedding vector. Extensive numerical experiments demonstrated that the proposed FedPCE approach generates comparable results to established personalization approaches, while drastically reducing the number of local parameters. This suggests that the adaptation problem is inherently low-dimensional. Moreover, the ability to fine-tune fewer parameters not only improves computational efficiency but also reduces the amount of labeled data required for effective adaptation. This is particularly important in the field of healthcare, where labeled data is usually scarce and very costly.

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
