# OpenReview forum: "FedPCE: Federated Personalized Client Embeddings"
_ICLR.cc/2025/Conference — ICLR 2025 Conference Withdrawn Submission_

### Official Review · Reviewer_Lue7 · 2024-10-17

**Soundness:** 2
**Presentation:** 1
**Contribution:** 1
**Rating:** 3
**Confidence:** 4

**Summary:**

The paper proposes FedPCE, a novel federated learning method that introduces personalized client embeddings to address challenges with non-IID data distributions among clients. FedPCE focuses on embedding vectors learned locally, which replace the traditional normalization layer parameters, making the adaptation to unseen clients more efficient. This method significantly reduces the number of parameters needed during fine-tuning, contributing to computational efficiency while maintaining competitive performance. The paper demonstrates the effectiveness of FedPCE across three datasets: CIFAR-10, CIFAR-100, and a Digits dataset.

**Strengths:**

- FedPCE introduces a novel use of local embeddings to reduce the need for parameter fine-tuning during client adaptation.
- FedPCE achieves comparable performance to traditional methods while fine-tuning only a fraction of the parameters.
- By focusing on personalized local embeddings, FedPCE is able to achieve improved client adaptation with fewer training samples, which is a crucial feature for FL applications in sensitive areas like healthcare.

**Weaknesses:**

- There is a need for an in-depth investigation into related research. The authors have not sufficiently investigated federated learning and domain adaptation studies. Above all, the paper only talks too general about each field of research. It isn't easy to understand the motivation of the study. FedBABU [1] proposed a method of not sharing the head layer, and it can be seen that there are some similarities with this study. MetaVers [2] proposed a personalized federated learning framework that leverages the embedding vectors of each client for personalization and requires no additional fine-tuning. FedAwS [3] also leverages the embedding vectors for federated learning.
- Comparisons with the latest federated learning frameworks are lacking.
- It is necessary to conduct experiments that include client participation rates. Full participation in federated learning is not realistic.
- The ablation study also requires comparison with other algorithms. It is necessary to prove that FedPCE is still valid even in the changed environments.
- The visibility of the main figure is not good. I recommend removing the gray background and modifying the overall figure.
- Overall, the contribution is trivial and lacks novelty.
- Real-world applicability discussion is needed.


[1] Oh, J., Kim, S., & Yun, S. Y. (2021). Fedbabu: Towards enhanced representation for federated image classification. arXiv preprint arXiv:2106.06042.

[2] Lim, J. H., Ha, S., & Yoon, S. W. (2024). MetaVers: Meta-Learned Versatile Representations for Personalized Federated Learning. In Proceedings of the IEEE/CVF Winter Conference on Applications of Computer Vision (pp. 2587-2596).

[3] Yu, F., Rawat, A. S., Menon, A., & Kumar, S. (2020, November). Federated learning with only positive labels. In International Conference on Machine Learning (pp. 10946-10956). PMLR.

**Questions:**

All described in Weakness.

---

> ### Author Response · Authors · 2024-11-14
>
> Dear Reviewer,
>
> Thank you for reviewing my paper and providing valuable feedback. I appreciate the time you put into reading it and your thoughtful suggestions. With your input, I hope I can improve my work.

---

### Official Review · Reviewer_w1H8 · 2024-10-19

**Soundness:** 2
**Presentation:** 3
**Contribution:** 2
**Rating:** 5
**Confidence:** 3

**Summary:**

The authors of the paper propose a method to address the challenge of non-IID (non-independent and identically distributed) data in federated learning (FL) by introducing Federated Personalized Client Embeddings (FedPCE). The method uses client-specific embeddings for the normalization layers, reducing the number of parameters that need to be fine-tuned during client adaptation.

**Strengths:**

1. The writing and structure of the paper is clear.
2. The proposed method is easy to understand.

**Weaknesses:**

1. The main problem is the effectiveness of the proposed method: FedPCE is outperformed by FedBN among many benchmark settings. FedPCE only outperforms FedBN when there are less training images used in different clients. The authors should possibly analyze the reason behind these results. Also, the differences between these two methods are not very clearly described in the paper, the authors should highlight more about this. Otherwise, the contribution of the paper might be limited to the community.
2. The adopted datasets are relatively simple. The authors might add more complex datasets such as TinyImageNet [1] or PACS [2].
3. The information provided in Figure 3 is not clear. The author should describe the experiment in more details.
4. The authors should provide a convergence rate analysis for both FedBN and FedPCE.

[1] Le, Ya, and Xuan Yang. "Tiny imagenet visual recognition challenge." CS 231N 7.7 (2015): 3.

[2] Li, Da, et al. "Deeper, broader and artier domain generalization." Proceedings of the IEEE international conference on computer vision. 2017.

**Questions:**

Please refer to the Weakness.

---

> ### Author Response · Authors · 2024-11-14
>
> Dear Reviewer,
>
> Thank you for reviewing my paper and providing feedback. I appreciate the time you put into reading it. With your input, I hope I can improve my work.

---

### Official Review · Reviewer_sgPj · 2024-10-30

**Soundness:** 3
**Presentation:** 3
**Contribution:** 2
**Rating:** 5
**Confidence:** 4

**Summary:**

The submission describes FedPCE, a method for personalized federated learning of the type where some parameters are shared among all clients and trained globally, and some are private to the clients and trained locally. Specifically, as in FedBN [Li et al, 2021], the private parameters influence only the normalization layers (scale and offset) of a deep network. The core difference of FedPCE to FedBN is that instead of learning scales and offsets explicitly, each client learns only a low-dimensional client descriptor for itself. The descriptor is used as input to another (small) network, the output of which are the actual normalization parameters. The secondary network is learned globally and jointly with the main network, so each client has much fewer private parameters to learn than in FedBN. Experiments are performed on standard datasets CIFAR10 and CIFAR100 as well as a mixture of digits datasets. They show that FedPCE often performs comparably to previous methods, and potentially offers an advantage if the amount of training data per client is very small.

**Strengths:**

Strengths:
- the problem studied (personalized federated learning) is timely and relevant
- the method described in the submission addresses is simple and makes sense
- the method is clearly described and the setting is mostly reproducible
- the experimental evaluation is mostly described well and appears fair towards the baselines

**Weaknesses:**

* the submisson does not present related prior work on hypernetwork-based personalized federated learning. As a consequence, it is hard to judge the technical novelty.

Specifically: some prior works exist in which a secondary network creates parameters for a primary network, with both networks being trained globally and the personalization consisting of a per-client descriptor. Typically, this is called a *hypernetwork* setup, e.g. [1,2,3]. The submission does differ from these works, e.g., in [1], the client descriptors are stored on the server, while [2] and [3] compute them on-the-fly from client data. Furthermore, in these work all parameters are predicted by the hypernetwork, not just the normalization layer parameters. Nevertheless, they constitute highly relevant prior work. The proposed method should be discussed in their context, and its novelty described accordingly.

[1] [Shamsian et al. "Personalized federated learning using hypernetworks", ICML 2021]
[2] [Amosy et al, "Inference-time personalized federated learning", https://openreview.net/forum?id=_DqUHcsQfaE 2022]
[3] [Scott et al, "PeFLL: Personalized Federated Learning by Learning to Learning", ICLR 2024]

A second branch of relevant work would be federated learning in which clients are clustered and per-cluster models are learned, e.g. [4,5]. This can be seen as a related setting where client descriptors are simply integers (cluster indicators).

[4] [Mansour et al. "Three approaches for personalization with applications to federated learning", https://arxiv.org/abs/2002.10619 2020]
[5] [Sattler et al, "Clustered federated learning: Model-agnostic distributed multi-task optimization under privacy constraints", IEEE TNNLS 2020]

* the "federated learning" aspect of the submission only involves the aspect of client data not being shared with other parties. Other aspects of federated learning, such as privacy of transmitted information, secure aggregation, client hardware heterogeneity, robustness to client dropout, are not addressed or discussed.

* some steps of the experimental evaluation are not fully clear, see "Questions" below

* the experimental results do not provide convincing reasons to adopt the new method

Specifically, the core experiment in the manuscript consists of a comparison of FedPCE to baselines in terms of the classification accuracy they achieve. However, in Tables 1 and 2 none of the method clearly outperforms the others, especially when taking into account the error intervals. FedBN and FedPCE performs comparably, and FedAvg (without personalization) is also quite competitive. I find it laudable that the authors openly present these numbers, but I find it hard to take them as a justification for adopting the new method over previous ones. In lack of better accuracy, the main difference between FedPCE and FedBN is that the former has to learn fewer private parameters. But this is not an obvious advantage. The private parameters are exactly the one which do *not* have to be transmitted via the network, there is no obvious gain in terms of communication cost. Later, the manuscript claims that the smaller number of parameters increases *computational* efficiency. But this is not obvious (as there is an additional network involved) and would need to be justified formally and/or experimentally.

The fact that FedAvg is competitive, at least on the Digits dataset, indicates that personalization might not actually be required here. Presumably, that is because the experiments are essentially a *covariate-shift* setting, where the distribution of inputs differs between clients, but a single consistent labeling function exists. With enough data and a rich enough network, FedAvg can learn this and perform well across the board. A second indication of this are Tables 3 and 5, which show that the embedding dimension and the width of the hypernetwork have rather small impact on FedPCE's performance (if any, given the rather large error interval and no test of statistical significance test is reported).

Presumably, a more suitable setting could be if clients distribution differ in their labeling function and little data is available per client. Figure 4 shows results at least about the latter, and indeed FedPCE shows promising in the very low data regime. However, the figure alone is insufficient to serve as justification for FedPCE. I believe the manuscript would benefit from a rewrite, in which the authors concentrate on the regime of per samples per clients (maybe with many clients). Also, the use of datasets with real non-iid client rather than just synthetic clients would add credibility to the results.

(Please note that I am not asking for new experiments to be reported in the author response, this is purely a suggestion for a revised manuscript, in case that occurs).

Additional comments:
* the manuscript should use \citep and \citet appropriately for citations.
* the reference section has some issues: currently it contains some arXiv version where published versions exist (e.g. Arivazhagan et al, Dumoulin et al, ...). Some references lack any publication venues (e.g. Kingma et al, Krizhevsky et al, Zhao et al). Spelling and capitalization are often inconsistent, e.g. of conference names.

**Questions:**

While I find the method to be interesting and promising, I have some questions the answer to which might influence my assessment of soundness, clarity and contribution. The scores I give to these and the overall rating are tentative so far, and I plan to revisit them after the author response.

* What are the technical differences of FedPCE to previous hypernetwork-based personalized FL methods (I believe I understand it, but it would be better to hear it phrased by the authors).

* Section 4.3 states that InstanceNorm is used instead of BatchNorm. Does this hold for the baselines, too? If yes, would FedBN maybe do better with BatchNorm, as it was originally proposed? If not, would the baselines maybe benefit from InstanceNorm, too? Same question also for the additional normalization layer in the classification head.

* How were all methods' hyperparameters chosen (e.g. learning rates/schedule for main network and embeddings, batchsize, number of local iterations, split of global/local aggregation, data augmentation)? Section 5.1 states that baselines and proposed method use same optimization hyperparameters. Where the same efforts spent to optimize the baselines for the tested setting as the proposed method?

---

> ### Author Response · Authors · 2024-11-14
>
> Dear Reviewer,
>
> Thank you for reviewing my paper and providing valuable feedback. I appreciate the time you put into reading it and your thoughtful suggestions. With your input, I hope I can improve my work.

---

### Official Review · Reviewer_DBh7 · 2024-11-04

**Soundness:** 1
**Presentation:** 1
**Contribution:** 1
**Rating:** 3
**Confidence:** 4

**Summary:**

In order to solve non-IID data distribution among clients issues in Federated Learning, this paper introduces Federated Personalized Client Embeddings method to utilize local embedding to capture the underlying structure of the normalization feature vectors.

**Strengths:**

No strength or meaningful point can be found in this paper.

**Weaknesses:**

- This paper is not well written. For example, in the “introduction” section, it is not clear why this paper would solve the non-IID problem and what is the main difference between existing approaches and this paper. In the “related works” section, reference papers are casually selected and presented. In the “method” section, many existing techniques are presented, which are unrelated to the main contributions.
- Experiments are not sufficient. Only small-scale datasets are used. Most results are worse or similar to three-year-old methods, e.g., FedBN.

**Questions:**

See weakness part

---

> ### Author Response · Authors · 2024-11-14
>
> Dear Reviewer,
>
> Thank you for reviewing my paper and providing feedback. I appreciate the time you put into reading it. With your input, I hope I can improve my work.

---

### Note · Authors · 2024-11-15

I have read and agree with the venue's withdrawal policy on behalf of myself and my co-authors.